# Artificial Bee Colony Algorithm with Adaptive Parameter Space Dimension: A Promising Tool for Geophysical Electromagnetic Induction Inversion

Dennis Wilken [1,*], Moritz Mercker [2,3], Peter Fischer [4], Andreas Vött [4], Ercan Erkul [1], Erica Corradini [1] and Natalie Pickartz [5]

1. Institute of Geosciences, Christian-Albrechts-University, 24118 Kiel, Germany; ercan.erkul@ifg.uni-kiel.de (E.E.); erica.corradini@ifg.uni-kiel.de (E.C.)
2. Bionum GmbH, Consultants in Biostatistics, 21129 Hamburg, Germany
3. Institute of Applied Mathematics, Heidelberg University, 69120 Heidelberg, Germany
4. Institute for Geography, Johannes Gutenberg-Universität, 55128 Mainz, Germany; p.fischer@geo.uni-mainz.de (P.F.)
5. State Office for Cultural Heritage Baden-Württemberg, 73728 Esslingen am Neckar, Germany
* Correspondence: dennis.wilken@ifg.uni-kiel.de

**Abstract:** Frequency-domain electromagnetic induction (FDEMI) methods are frequently used in non-invasive, area-wise mapping of the subsurface electromagnetic soil properties. A crucial part of data analysis is the geophysical inversion of the data, resulting in either conductivity and/or magnetic susceptibility subsurface distributions. We present a novel 1D stochastic optimization approach that combines dimension-adapting reversible jump Markov chain Monte Carlo (MCMC) with artificial bee colony (ABC) optimization for geophysical inversion, with specific application to frequency-domain electromagnetic induction (FDEMI) data. Several solution models of simplified model geometry and a variable number of model knots, which are found by the inversion method, are used to create re-sampled resulting average models. We present synthetic test inversions using conductivity models based on 14 direct-push (DP) EC logs from Greece, Italy, and Germany, as well as field data applications using multi-coil FDEMI devices from three sites in Azerbaijan and Germany. These examples show that the method can effectively lead to solutions that resemble the known DP input models or image reasonable stratigraphic and archaeological features in the field data. Neighboring 1D solutions on field data examples show high coherence along profiles even though each 1D inversion is independently handled. The computational effort for one 1D inversion is less than 120,000 forward calculations, which is much less than usually needed in MCMC inversions, whereas the resulting models show more plausible solutions due to the dimension-adapting properties of the inversion method.

**Keywords:** FDEMI; inversion; artificial bee colony; Markov chain Monte Carlo; electromagnetic induction; geophysics

## 1. Introduction

Electromagnetic induction (EMI) measurements allow non-invasive area-wise mapping of the subsurface electrical conductivity (EC) and magnetic permeability. Together with the capability of providing depth information, this makes them an important and widely used tool in geophysical exploration (e.g., [1]). In frequency-domain EMI (FDEMI), a controlled source electromagnetic method (CSEM), a transmitter coil emits a primary harmonic oscillating electromagnetic field (kHz frequency range). This field induces eddy currents in the subsoil that generate a secondary field, which in superposition with the primary field, is recorded at the receiver coils. FDEMI measurements are thus determined by conductivity, the magnetic permeability of the ground, and depending on the used frequency, also dielectric permittivity (e.g., [2]). Different depth sensitivities can be achieved

by using multiple receiver coils of different distances to the emitter coil, different coil orientations, varying heights above ground, or different signal frequencies. Hereby, vertical coplanar (VCP, both coil planes perpendicular to surface) orientations are more sensitive to the shallow subsurface, and measurements made in horizontal coplanar (HCP, both coil planes parallel to surface) orientation are sensitive to larger depths, respectively. Larger coil distances and lower frequencies also enable larger sounding depths (see e.g., [3]). In the past, several applications of FDEMI measurements have been published and the technique of FDEMI devices and analysis has been significantly improved. These developments allow surveying large areas efficiently (e.g., [4]).

An important part in data analysis and the step from maps of apparent conductivity to specific conductivity distribution models is geophysical inversion. Geophysical inversion is the inverse procedure of the forward model which calculates data from a given subsurface conductivity distribution. The inverse problem usually fits synthetic forward modelled data, based on a variable subsurface model of the desired physical subsoil parameter, to measured data points, yielding a subsurface model of optimal data fit. If the physical relation between data and subsurface model is linear, this can be done via a matrix inversion (see e.g., [5]). In the more common non-linear case, a linearization can be done and the data-fitting optimization problem can be solved iteratively using gradient search and regularization. However, in many cases the optimization problems are highly non-unique, making the application of global searching probabilistic methods worthwhile.

Several works in different geophysical applications have conducted so (e.g., [6–8]). Furthermore, probabilistic methods allow a proper interpretation of model uncertainties and model parameter trade-offs based on statistical analysis (e.g., [9]). However, these methods need a large amount of forward calculations, making the efficiency of such a method a very important factor. In terms of FDEMI inversion, several algorithms have been published and tested in the past. These methods can recover either only subsurface electrical conductivity or, in cases where applicable, magnetic permeability and dielectric permittivity. Most FDEMI inversion algorithms solve for 1D layered-earth models. Ref. [10] summarized that most works use non-linear least-squares gradient algorithms (e.g., [11–15]). Although these algorithms provide fast convergence, an analysis of parameter uncertainty, correlation, and non-uniqueness is often left unaddressed ([10]). Analysis of uncertainty within the least-squares framework is typically performed through the use of measures computed from the linearized Jacobian, such as the posterior covariance or resolution matrices, but limited to its diagonal elements ([10]), thus not addressing trade-offs. Because of this disadvantage, several probabilistic optimization methods have been applied to the problem. Ref. [10] introduced a Bayesian Markov chain Monte Carlo (MCMC) approach to provide a direct assessment of parameter uncertainty, correlation, and non-uniqueness/ambiguity. This approach for the first time allowed a variable number of layers in the inverted subsurface models: This means that the dimension of each model was an inversion parameter, too. Ref. [16] combined a global grid search with a local simplex algorithm. Refs. [3,17,18] introduced different MCMC algorithms for EMI inversion, whilst in [19], a Bayesian sampling with dimensionality reduction technique was implemented, making the step to neural network methods in EMI inversion. Ref. [20] applied a Kalman ensemble generator to the one-dimensional probabilistic multi-layer inversion of EMI data to derive conductivity and susceptibility simultaneously. Ref. [21] used the shuffled complex evolution (SCE) algorithm. In several publications on geophysical inversion, the effectiveness and benefits of probabilistic population-based (evolutionary, genetic algorithms, and swarm intelligence optimization) methods are shown (e.g., [6–8,22,23]). This family of optimization methods combines global search, trade-off, and uncertainty analysis, as well as relatively fast convergence properties. Thus, it is obvious to use them in EMI inversion. First approaches have been using particle swarm optimization (PSO) (e.g., [24]).

The presented work deals with a novel stochastic optimization approach in combining artificial bee colony (ABC) with reversible jump Markov chain Monte Carlo (RJ-MCMC) optimization for geophysical inversion, with specific application to FDEMI data. In par-

ticular, our general approach follows and extends the work of [25]. They showed for the case of seismic traveltime tomography how several RJ-MCMC inversions of simplified model geometry and a variable number of model knots can be used to create a re-sampled resulting average model. Nevertheless, the RJ-MCMC approach needs a large amount of forward calculations to derive this solution model. In this paper, we thus pose the following questions based on this state of research:

- Is it possible to upgrade a swarm intelligence optimization approach with dimension-adapting properties as used in the MCMC approach by [10] or [25]?
- How does a hybrid approach perform, having available both swarm intelligence convergence effectiveness as well as the Bayesian-statistics-guided dimension-adapting properties of the RJ-MCMC approach?

To answer these questions, we tested and optimized our approach on the usually highly under-determined problem of FDEMI inversion. Such data sets enable us to discuss how and why complex 1D underground models can be derived from multiple ambiguous solutions to fit a limited number of measurements. Finally, we applied the resulting method to three field data sets, which enabled us to evaluate the result based on the variance distribution and coherence along the profiles.

The paper is set up as follows. We will first introduce the forward model and our choice of model parametrization. Subsequently, we introduce the used inversion scheme and the statistical solution-model estimation. Then, we present fourteen test models, based on direct-push EC-log data from six different sites. These models represent a large number of possible subsurface situations at different sites across Europe. Finally, the field application examples are introduced. After presenting the results, we close with a discussion and a conclusion.

## 2. Methods and Data

In this section, we will describe the methodology behind our used forward model, the way of 1D conductivity–depth model discretization and model-averaging approach, and the choice of our test models and parameter space, and describe the ABC and MCMC combined optimization approach.

### 2.1. The Forward Model

For a given layered conductivity earth model, the electromagnetic response can be calculated in different ways. A certain degree of approximation can be applied depending on the so-called induction number

$$B = s \cdot \sqrt{\pi f \mu_0 \sigma}, \tag{1}$$

with $s[m]$ being the coil distance, $\sigma[S/m]$ the electrical conductivity of the subsoil, $\mu_0[N \cdot A^{-2}]$ the vacuum magnetic permeability, and $f[Hz]$ the signal frequency. If HCP and VCP coil orientations are measured and $B << 1$, meaning coil distance is much smaller than skin depth ($s/B$), which is called the LIN (low induction number) case, the forward model can be simplified by an approximation using cumulative sensitivity functions introduced by [26] to obtain the quadrature component. This component is the imaginary part of the measured ratio of primary and secondary magnetic field. We will refer to this as the LIN forward model in this paper. To keep it simple and fast, in testing the capabilities of the presented inversion approach, we used this LIN approximation as a forward model for the synthetic tests in this work. Nevertheless, the inversion program used in this paper also comprises forward calculation by [27], using full problem solutions of Maxwell's equations based on indefinite integral solution involving recursive-layer-based reflection coefficient determination. This forward calculation option can be chosen if the LIN approximation is not valid. We will refer to this as the full forward model throughout this paper.

### 2.2. Model Parametrization and Error Estimate

The question of model discretization plays a significant role in the regarded inversion problem. A model usually consists of a layered half-space having a fixed number of layers, each defined by their thickness $h$ and conductivity $\sigma$ (and if regarded the magnetic permeability). On the one hand, models need to have as few parameters as possible to reduce ambiguities due to the sparse inverse problem. On the other hand, the subsoil usually comprises a smooth conductivity–depth solution that would need to be described by a large number of homogeneous layers or gradient layers, making the problem even more sparse. An often cited solution for this problem is the work by [28], called Occam's Inversion, which uses the smoothest (simplest) model to describe the data with a given tolerance. In this work, we combine these two ideas. The basic models are parameterized as follows, using a simple constant-layer approach. A model is defined as

$$\vec{m} = (D, z_1, \ldots z_D, \sigma_1, \ldots, \sigma_D), \tag{2}$$

where $z_i$ is the depth of the center of layer $i$ in $m$ and $\sigma_i$ the corresponding conductivity $(S/m)$. $D$ is the dimension of the model. The conductivity–depth distribution based on these $(z_i, \sigma_i)$-points is generated by a 1D Voronoi rule. Layer interfaces are thus half between two $z_i$. $D$ will vary during the inversion process within a reasonable range, based on the number of data points available. The inversion process results in a set of these very simple but well-fitting models (see examples in Figure 1). Each individual model is coarse and blocky and usually does not represent the subsoil situation very well. Thus, instead of being forced to make a choice between these simple solutions, an average solution model can be defined that represents all found solutions, weighted by their misfit. For this purpose, the simple solution models are re-sampled with a finer depth spacing $dz$ ($\sigma(z) = \sigma(i \cdot dz)$, $i = 0 : Nz$, $Nz = z_{max}/dz$). From these re-sampled models, statistical properties like the mean and standard deviation can be derived for each sublayer (see Figure 1). During the stochastic inversion process, numerous models and misfit values are saved and can be used for the calculation of these average/expected model and standard deviation estimates. For this purpose, [25] analyzed the re-sampled models to derive the arithmetic mean, standard deviation, or median. They treat the average model as the reference solution, and the standard deviation is interpreted as a measure of the model error. In our case of 1D FDEMI inversion, we chose to calculate the expected solution model and model error estimation following a more general statistical analysis for stochastic inversion, as used for example in [7] or [9]. We use the $l = 1, \ldots, n_b$ best models and perform the following steps:

1. Define an evenly finer discretization with depth $z_i = 0, dz, 2dz, \ldots, z_{max}$.
2. Discretize all $\vec{m}_l$ according to $z_i$. This leads to finer discretized models $\vec{M}_l$.
3. Calculate the expected model using

$$< \vec{M} > = \frac{1}{N} \sum_{l=1}^{n_b} \vec{M}_l \xi(\vec{M}_l), \tag{3}$$

with $N$ being the partition function approximated by

$$N = \sum_{l=1}^{n_b} e^{-Q(\vec{m}_l)} \tag{4}$$

and $\xi$ being the probability density function based on the quality value

$$\xi(\vec{M}_l) = e^{-Q(\vec{m}_l)}, \tag{5}$$

where $Q$ is the quality function measuring the misfit between measured and modeled data (see next section). Variances are then the diagonal elements of the covariance matrix

$$C = \frac{1}{N} \sum_{l=1}^{n_b} (\vec{M}_l - < \vec{M} >)(\vec{M}_l - < \vec{M} >)^T \xi(\vec{M}_l). \tag{6}$$

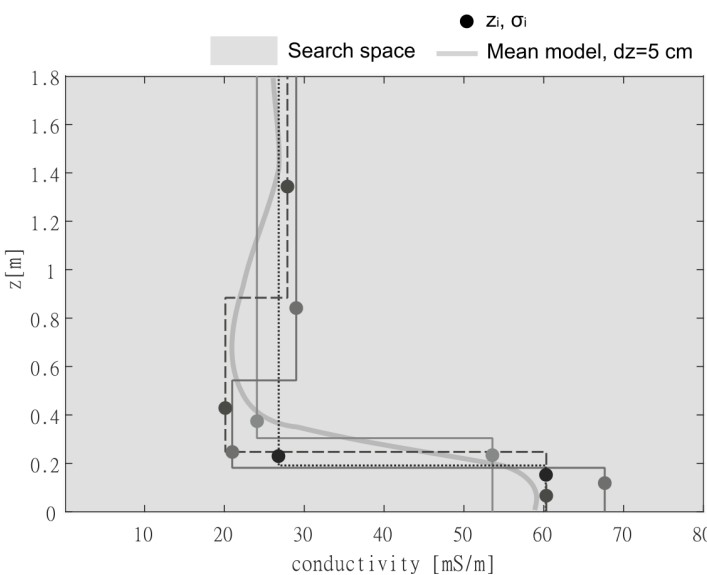

**Figure 1.** Parametrization of models. Dots indicate the Voronoi centers of constant layers. The simple solution models calculated from these Voronoi centers are plotted in thin gray, solid, dotted and dashed lines. Above, an exemplary re-sampled average model is shown (light grey, thick line).

### 2.3. The Inverse Problem

Based on the above-defined model $\vec{m}$ and the forward calculation by [26] or [27], we can define the inverse problem as a least-squares problem with a quality function following [21], weighted by normalized error values:

$$Q(\vec{m}) = \frac{1}{M} \sum_{i=1}^{M} \frac{\Delta\sigma_{min}}{\Delta\sigma_i} \frac{|d_i^{meas} - d_i^{mod}(\vec{m})|^p}{|d_i^{meas}|^p}, \tag{7}$$

where $M$ is the number of data points/coil distances and orientations; $d_i^{meas}$ are the measured data (imaginary parts of magnetic fields or apparent conductivity (either logarithmic or linear values), depending on full- or LIN forward solutions); $d_i^{mod}(\vec{m})$ are the modeled data based on the subsurface model $\vec{m}$; and $p$ denotes the norm used. In our case, we tested $p = 1, 2$. $\Delta\sigma_{min}$ and $\Delta\sigma_i$ are the minimum and actual error estimates of the used data points. The resulting task is to minimize $Q(\vec{m})$.

### 2.4. Optimization Approach

As most stochastic methods like evolutionary algorithms/strategies (e.g., [29]), the neighborhood algorithm (NA [30]), or particle swarm optimization (PSO) and its variants (e.g., [7]) use binary operations in parameter space to recombine or move particles or individuals or agents, a dimension-adapting strategy is thus impossible to include in these algorithms. Different particles of different dimensions cannot be combined and have no unique distance measure. All these algorithms are thus not suitable to be used as a global searcher if the dimension of a problem is a parameter, too. In analogy to the standard particle swarm optimization, which adapts the behavior of swarms of fish or birds, [31] developed a method based on the behavior of a bee hive. This method offers several recombination methods that solve the above-mentioned problem. The artificial bee colony algorithm (ABC) is basically made of several local searches, supported by a hive collective, only connected by a change in one randomly chosen parameter dimension. Thus, ABC enables us to change the dimension of each local search and offers a global search feature due to the guided abandonment of the local searches (see, e.g., [7]).

The driving force behind the movement of a bee swarm is the search for food sources and the effective exploitation of those. A fundamental model for simulating the collective intelligence of a bee hive needs three components ([31]). These are the food sources them-

selves, the worker drones employed at a source and exploiting the source (local search), and drones that are not permanently assigned to a source (enhanced local search). Two basic actions are also required: assigning worker drones to a food source and leaving a depleted source. In analogy to an optimization problem, a possible food source would correspond to a region of the parameter space. The area is scanned by the drones, and the yield of this source is determined by the value of the local quality function and the success of the optimization. The employed drones are thus particles that carry out a stochastic search in such an area, harvesting the area, so to speak. Information about the processed food source is shared with the hive in order to place drone reinforcements at food sources with a certain probability. Unemployed drones have the task of looking for possible sources of food. There are two types of unemployed drones: drones that randomly explore new areas of the parameter space after the depletion of a food source and drones that use the information from the worker drones to choose a known source and work there. For the ABC algorithm, this means that the bee colony consists of three types of particles: employed drones, helping drones, and exploration drones. The employed drones will look for a food source and exploit it, i.e., conduct a local search in an area of the parameter space. They share information about that source in the swarm, and the helping drones choose a source to work with. An employed bee whose food source is empty (no convergence progress) becomes an exploration drone and randomly finds a new source. Our adapted ABC approach combines ABC, used as a frame algorithm, and MCMC, used as local search. Each bee at a food source performs an MCMC search, using the birth/death algorithm also used in [10,25].

The algorithm comprises the following steps:

1.  Define the size of the bee hive, consisting of $n$ employed and $n$ helping bees. The swarm has a total size of $2 \cdot n$.

2.  Initialize all $j = 1, \ldots, n$ employed bees, which means randomly generate each bee as a position in search space:

$$\vec{m}_j = (D_j, z_1, z_2, z_3, \ldots, z_i, \ldots, z_D, \sigma_1, \sigma_2, \sigma_3, \ldots, \sigma_i, \ldots, \sigma_D) \tag{8}$$

with $D_j$ being the number of model knots, including, $z_i$ depths and $\sigma_i$ conductivities ($i = 1, \ldots, D_j$), as described in the section Model Parametrization and Error Estimate (Section 2.2). The positions of these bees reflect the position of the so-called food sources.

3.  Evaluate the value of each food source by calculating the quality function $Q(\vec{m}_j)$ for $j = 1 \ldots n$.

4.  Calculate an assignment probability $P_j$ for each food source $j$, depending on its quality value:

$$P_j = \frac{Q_{max} - Q_j}{\sum_{i=1}^{n}(Q_{max} - Q_i)}, \tag{9}$$

with $Q_{max}$ being the actual maximum of all $Q$.

5.  Create $c = 1, \ldots, n$ helping bees. For each helping bee, $\vec{m}^c$ choose an existing food source $\vec{m}^k$ based on their probabilities $P$, and create the helping bee by a search step around the employed bee's position (food source $k$). Following [31], this is achieved by altering the value of the employed bee in a randomly chosen dimension $i$ in the direction of another randomly chosen food source $\vec{m}^l$:

$$m_i^c = m_i^k + (2r - 1)(m_i^k - m_i^l), \tag{10}$$

with $r$ being a uniform random number in $0, 1$. Considering that each food source stands alone as a search spot, and several helping bees can be assigned to a food source. This step can be seen as a local random search, although connected to the directions and probabilities of the other food sources. If so, the dimensionality of each food source can change without altering the algorithm's basics.

6. Calculate the minimum quality value of all employed and helping bees, $Q_{best} = min(Q_c, Q_j)$, $c = 1 \dots n$, $j = 1 \dots n$, to measure convergence.

7. Begin the main iteration loop:

   (a) Change employed bees. For each employed bee $j$, decide randomly between two possible steps with a probability of 0.5:

   - Perform ABC local search: Change the value of the employed bee $\vec{m}^j$ in a randomly chosen dimension $i$ in the direction of another randomly chosen food source $\vec{m}^k$ (k is a natural random number out of $1, n$):

   $$m_i^j = m_i^j + (2r - 1)(m_i^j - m_i^k),\qquad(11)$$

   with r being a uniform random number in $0, 1$. The dimension $i$ is randomly chosen up to the minimum of $D_k, D_j$.

   - Perform the RJ-MCMC step to create a proposed bee. Randomly choose between two possibilities:

   **I.birth**: Add a new model point $(z, \sigma)$ after a randomly chosen model point into the employed bee (increase $D_j$ by 1), or

   **II.death**: Remove a randomly chosen model point from the employed bee (reduce $D_j$ by 1).

   Whether a proposed bee will be accepted is based on the proposal distribution being the product of quotients of posterior probability of the proposed, $\vec{m}'$, and the original model, $\vec{m}$, given the data $\vec{d}$ and the proposal distribution of the original model given the proposed model and vice versa ([10]):

   $$\alpha = min\left[1, \frac{p(\vec{m}'|\vec{d})}{p(\vec{m}|\vec{d})}\frac{p(\vec{m}|\vec{m}')}{p(\vec{m}'|\vec{m})}\right].\qquad(12)$$

   For a birth step, the probability of acceptance simplifies to

   $$\alpha(\vec{m}'|\vec{m}) = min\left[1, \frac{\Delta\sigma}{\delta\sigma_2\sqrt{2\pi}}exp\left(\frac{(\sigma' - \sigma)^2}{2 \cdot \delta\sigma_2^2} - \frac{Q(\vec{m}') - Q(\vec{m})}{2}\right)\right],\qquad(13)$$

   with $\Delta\sigma = \sigma_{max} - \sigma_{min}$, $\delta\sigma_2$ being the prior covariance and $\sigma'$ being the conductivity of the newly inserted model conductivity. $\sigma$ is the interpolated conductivity of the original model at the depth of the new model point. For a death step, the probability is given as

   $$\alpha(\vec{m}'|\vec{m}) = min\left[1, \frac{\delta\sigma_2\sqrt{2\pi}}{\Delta\sigma}exp\left(\frac{(\sigma' - \sigma)^2}{2 \cdot \delta\sigma_2^2} - \frac{Q(\vec{m}') - Q(\vec{m})}{2}\right)\right],\qquad(14)$$

   with $\sigma'$ being the average conductivity at the depth of the removed model point and $\sigma$ being the conductivity of the removed model point. $\alpha$ becomes 0 if the proposed conductivities are outside of the parameter space or the dimension exceeds given limits.

   (b) Evaluate and save all new quality values of the employed bees, and re-calculate the assignment probabilities $P_j$.

   (c) Check for stagnation of all food sources by checking if the change of quality value

   $$dQ_j = (Q_j - Q_{j,old})/Q_j\qquad(15)$$

   is less then a constant $\epsilon_1$ for more than $dk$ iterations. If so, the food source is abandoned, and the employed bee performs a random reset in the next iteration.

(d)     Randomly re-assign helping bees based on probabilities $P_j$. Alter their position by using the rules in 7 (a) but with a third choice of performing no death/birth step.

(e)     Evaluate and save all missing quality values of the helping bees.

8.    Update $Q_{best}$.

9.    Check the stopping criterion, which is either the maximum number of iterations reached or $Q_{best}$ has fallen below a constant $\epsilon_2$. Otherwise, go to 7.

### 2.5. Dp Test Models

To initially test our inversion scheme, we use fourteen direct-push (DP) electrical conductivity (EC)-downhole logs representing the in situ subsoil conductivity at seven different sites across Europe. The conductivity–depth curves are used to calculate the FDEMI response in the HCP and VCP modes, which will then be inverted using the proposed inversion scheme. The data sets used can be found in the literature, sorted by their site:

1.    TEV ([32]) is an example from an abandoned Tiber meander (Italy), comprising fluvial deposits of different Tiber channel generations.

2.    BIE ([33]) is an example from Biersdorf in the Eiffel area in Germany (Rhenish Massif, Rheinland-Pfalz), representing hillslope debris flow sediments.

3.    DUV ([34]) is an example from the Duvensee bog (Germany), comprising low-conductive glacial sand and layers of different Gyttja sediments.

4.    KAI ([35]) is an example from the Kaiafa lagoon located at the western Peloponnese in Greece, comprising mostly allochthonous sand sheets.

5.    REM ([36]) is an example from a Loess–Palaeosol sequence (LPS) in the Middle Rhine Valley, Germany (Schwalbenberg LPS).

6.    TRE ([37]) is an example from the Wadden Sea area of northern Germany (North Frisia), comprising mainly sandy, silty and organic layers from tidal flats and marshlands.

7.    AST is an example from an ancient Roman artificial channel site in Hesse (Germany) (see also the section on field data applications).

All EC logs were derived using a Geoprobe 540 MO system mounted to a Nordmeyer drill rig in combination with a Geoprobe K6050 HPT probe or a Geoprobe SC400 Conductivity Wenner EC Probe (Geoprobe Environmental Technologies S.A., Braine-le-Comte, Belgium). The subsoil conductivity curves for all fourteen examples can be seen in Figure 2. Here, not only are the EC logs plotted, but also the spatial-wavelength power spectra of the data. The spectra show that the EC logs show small-scale layers in a centimeter range but also a dominating, long-wavelength portion of features above 2 m in size. Looking at this long-wavelength portion, which probably can be resolved with FDEMI devices, we can see that in these cases, the first few meters of subsoil can be represented in terms of only a few layers.

To generate synthetic FDEMI data, we use these logs for FDEMI forward modeling. Therefore, the EC logs were re-sampled to constant 0.1 m thick sub-layers. The model's maximum depth was chosen based on the sensitivity properties of one FDEMI device, the CMD Explorer (GF Instruments, Brno, Czech Republic), which we chose as reference for this test study. We thus have a data set of three HCP, three VCP, or six HCP and VCP measures for the three different coil distances of the EMI device (1.48 m, 2.82 m, and 4.49 m (see [38])), giving an estimate of the maximum penetration of about 6.5 m in the HCP case. The device measures at a frequency of 10 kHz and is considered to be carried at ground level in this case.

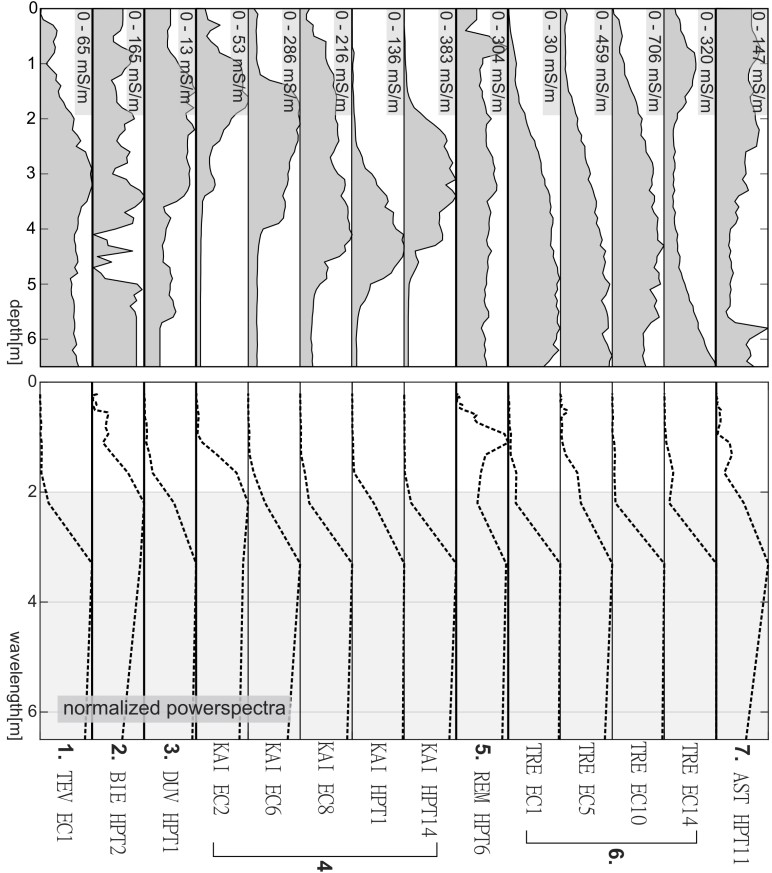

**Figure 2.** (**Top**) Direct-push EC logs, determined at fourteen different sites. (**Bottom**) Corresponding power spectra of the EC curves.

The inversion parameters were chosen based on test runs, resulting in best performance with $n = 400$, $D_{min} = 2$, $D_{max} = 4$, maximum number of iterations of 200, stagnation change threshold for food sources $\epsilon_1 = 1.0 \times 10^{-4}$ mS/m, and general stopping criterion $\epsilon_2 = 1.0 \times 10^{-3}$ mS/m. The 300 best (based on their misfit value) models found by each inversion were saved for further analysis. By calculating the RMS error between the resulting, re-sampled expected models and the known EC-log test models, we were able to evaluate some of the used inversion parameters in a grid search. This included $D_{max}$, $dk$, the choice of $L_1$ or $L_2$ norm, and $n_p$, the number of best models to calculate expected model and variance. These tests showed that for the HCP and VCP joint-inversion case, $D_{max} = 4$ (which is a natural choice in terms of number of measurements), $dk = 5$, the $L_2$ norm, and $n_p = 30$ create the best fit. The search space of the inversion was set to one-fourth of the minimum conductivity of a data set up to twice the maximum conductivity for all test examples. The MCMC uses $\delta\sigma_2 = 0.68 \times (\sigma_{max} - \sigma_{min})$ as an assumption for the prior covariance.

### 2.6. Field Datasets

We demonstrate the application of our inversion strategy on three example profiles from archaeo-geophysical field data sets measured with the CMD Miniexplorer or the CMD Explorer (GF Instruments, Brno, Czech Republic). The choice of examples was based on different additional information available. The first data set example was chosen because a small-scale structure, a tomb, is known to be on the profile, allowing us to test the ability of the inversion to image such small structures. The second example was chosen due to the availability of a direct-push EC-log data, allowing a ground truth of the inversion. The third data set was chosen because of its high dynamic range in conductivity of one order

of magnitude, allowing ua to test the behavior of the inversion and the coherence of the results with self-adapting parameter space.

The measurement setups for the three data sets are shown in Figure 3. All data sets are measured only in HCP mode, with a sampling interval of 0.1 s. Acquisition was carried out in continuous mode, meaning the device was moved along the profile continuously while measuring GPS position and data, as shown in Figure 3. To summarize the application examples:

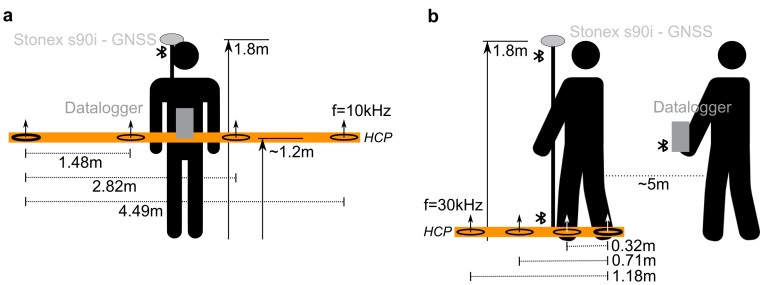

**Figure 3.** Acquisition geometries for (**a**) GF Instruments CMD Explorer and (**b**) GF Instruments CMD Miniexplorer.

Example 1: Measured with the CMD Miniexplorer in the Kurgan (burial mount) area on the Uzun–Rama plateau in central Azerbaijan. These mounts were constructed and used from the mid-4th to 1st millennium BC. For details of the site, see [39]. The example profile comprises 204 single independent 1D inversions. The LIN forward model was chosen due to the low apparent conductivity along the profiles ranging from 4 mS/m to 13 mS/m. For measurement parameters and setup, see Figure 3b.

Example 2: Measured with the CMD Explorer at a Roman river fortlet (burgus) site in Hesse (Germany). During the 1st century AD, the Romans performed several river alterations in the vicinity of the River Rhine in Hesse, including channels, holding anchoring sites protected by these fortlets. The regarded example profiles cross such a channel. In the middle of the assumed channel, a direct-push EC log was performed alongside a hydraulic profiling tool, using a Geoprobe 540 MO system mounted to a Nordmeyer drill rig in combination with a Geoprobe K6050 HPT probe. The example profile comprises 40 single independent 1D inversions. The full solution forward model by [27] was used. For measurement parameters and setup, see Figure 3a.

Example 3: Measured with the CMD Explorer at the edge of a preboreal lake site at Duvensee (see, e.g., [34]). The site is at the western sandy-loamy shore of the former Duvensee lake. The lake itself today is silted up mainly with peat and gyttja layers (mud of organic origin deposited in lakes and bogs). As a reference data set, a GPR profile was performed along the profile, imaging at least the first two to three meters of subsoil. The profile was recorded with a GSSI SIR-4000 system and a GSSI 200 MHz antenna. Processing included constant trace distance of 2 cm, amplitude offset removal, $t_0$ correction (14 ns), band-pass filter opening at 50 MHz and closing at 400 MHz, time-gain function, and finally, a topographic migration with a constant velocity of 7.2 cm/ns derived from hyperbola fitting. The EMI example profile comprises 79 single independent 1D inversions. The full solution forward model by [27] was used. Each 1D inversion used an automatically adapted search space in terms of conductivity ranging from one tenth of the minimum apparent conductivity to twice the maximum conductivity. For measurement parameters and setup, see Figure 3a.

Inversion parameters were $n = 400$, $D_{min} = 2$, $D_{max} = 4$, maximum number of iterations of 200, stagnation threshold for food sources $\epsilon_1 = 1.0 \times 10^{-4}$ mS/m, and $dk = 3$. Misfit

calculation was performed using the L1 norm. The re-sampling of 1D models was in steps of 5 cm for the CMD Miniexplorer case and 20 cm for the CMD Explorer cases.

## 3. Results

### 3.1. DP Test Models

Figures 4 and 5 show the results of the direct-push test model inversions in a direct comparison to the true-input conductivity–depth models. The inversion results are shown in terms of the expected model and one $\sigma$ (68%) confidence interval (blue and green areas) for both cases, first for six data points (measured in HCP and VCP orientation) (green solid line) and second for three data points, measured only in HCP orientation (blue dashed line). All models are plotted together with their derived normalized covariance matrix, $\tilde{C}_{i,j} = \frac{C_{i,j}}{\sqrt{C_{i,i}C_{j,j}}}$. As $\tilde{C}$ is symmetrical, only one-half is displayed. Variances are the diagonal elements, whereas the other elements represent covariances and thus allow a discussion of trade-off effects for a certain model. The values close to the main diagonal illustrate local trade-offs between thin layers and are thus a measure of resolution limitations (a broader diagonal thus shows less resolved models). Elements further off the diagonal show trade-offs between different separate layers. Some of the examples in Figures 4 and 5 have such exemplary areas highlighted by dashed boxes, labeled with capital letters. These letters can also be found in the corresponding depth ranges of the resulting models. All results show the typical effects due to the limitation and ambiguity of the measuring method itself. For example, most of the thin layers (below 2 m in thickness) cannot be resolved. This is especially obvious in model BIE-HPT2 (Figure 4e). Equivalent layer effects occur as expected (see also, e.g., in models DUV-HPT1 in Figure 4b and TRE-EC1 and 10 in Figure 5b,c). Uncertainties and trade-off effects increase if short-wavelength changes are in the same amplitude range as long-wavelength trends (e.g., REM-HPT6 (Figure 5e), BIE-HPT2). In general, the dominant long-wavelength trend of the models could be reconstructed for most models with a simpler long-wavelength geometry, meaning a two- or three-layer case, independent of conductivity range.

The model AST-HPT11 (Figure 5d) needs to be treated separately because it is investigated in two ways: first as a DP test model and second as an unknown model being part of the second EMI field data example. At this point, it is noteworthy that in Figure 5d, a trade-off with a deeper high-conductive layer is observed (labeled A). This trade-off does not appear in the EMI inversion of the corresponding field data set, as shown later. This is due to the fact that the EMI device was carried at 1 m height, reducing the overall depth range, and thus not reaching this high conductivity layer at about 6 m depth.

To test the influence of noise on the inversion results, the test model TRE-EC10 was tested with added Gaussian noise in the range of two, four, and ten times the data standard deviation. The noise on the data results from the instrument accuracy and the measurement setup that is suffering from noise by movement. The standard deviation of the data was estimated by using a field test, measuring 1000 data points with Field Setup (a) (Figure 3) on one spot while pretending movement. The data of the second coil distance showed the largest standard deviation (see Figure 6, top left), which was then chosen as the representative value for setup/instrument noise. Figure 6 shows the inversion results for all three noise levels.

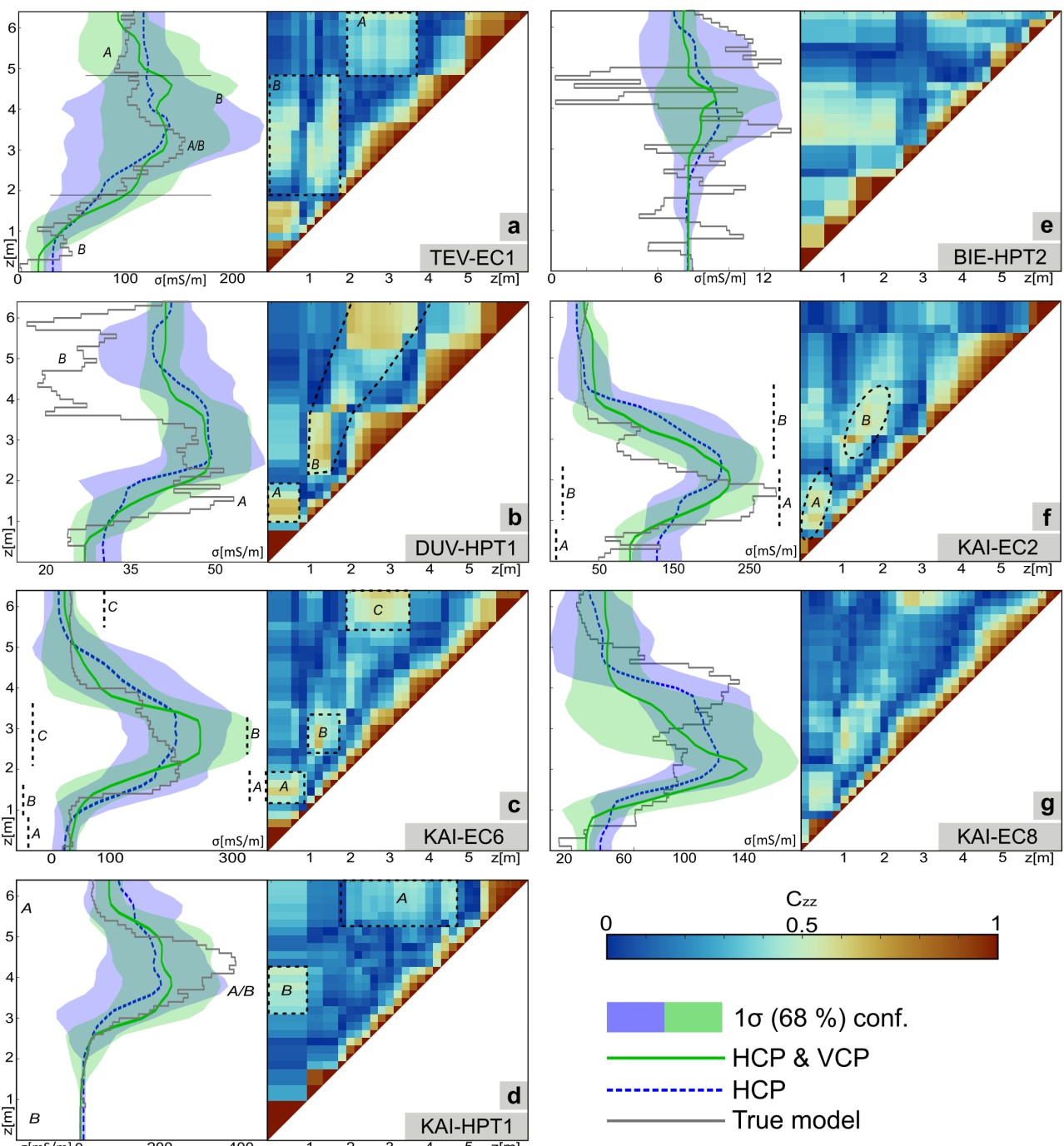

**Figure 4.** Results of DP test models, Part 1: first 7 of the 14 DP test models (**a**–**g**) with original conductivity-depth profile (gray curve), inversion result in terms of expected model for six data points in HCP and VCP orientation (green solid line) and for three data points only in HCP orientation (blue dashed line). The blue and green areas indicate the one $\sigma$ (68%) confidence interval. The gray line represents the true conductivity model. On the right side of each inversion result, the derived covariance matrix is displayed. Labels A and B in (**a**,**b**,**c**,**d**,**f**) refer to areas of high covariance and are discussed in the text.

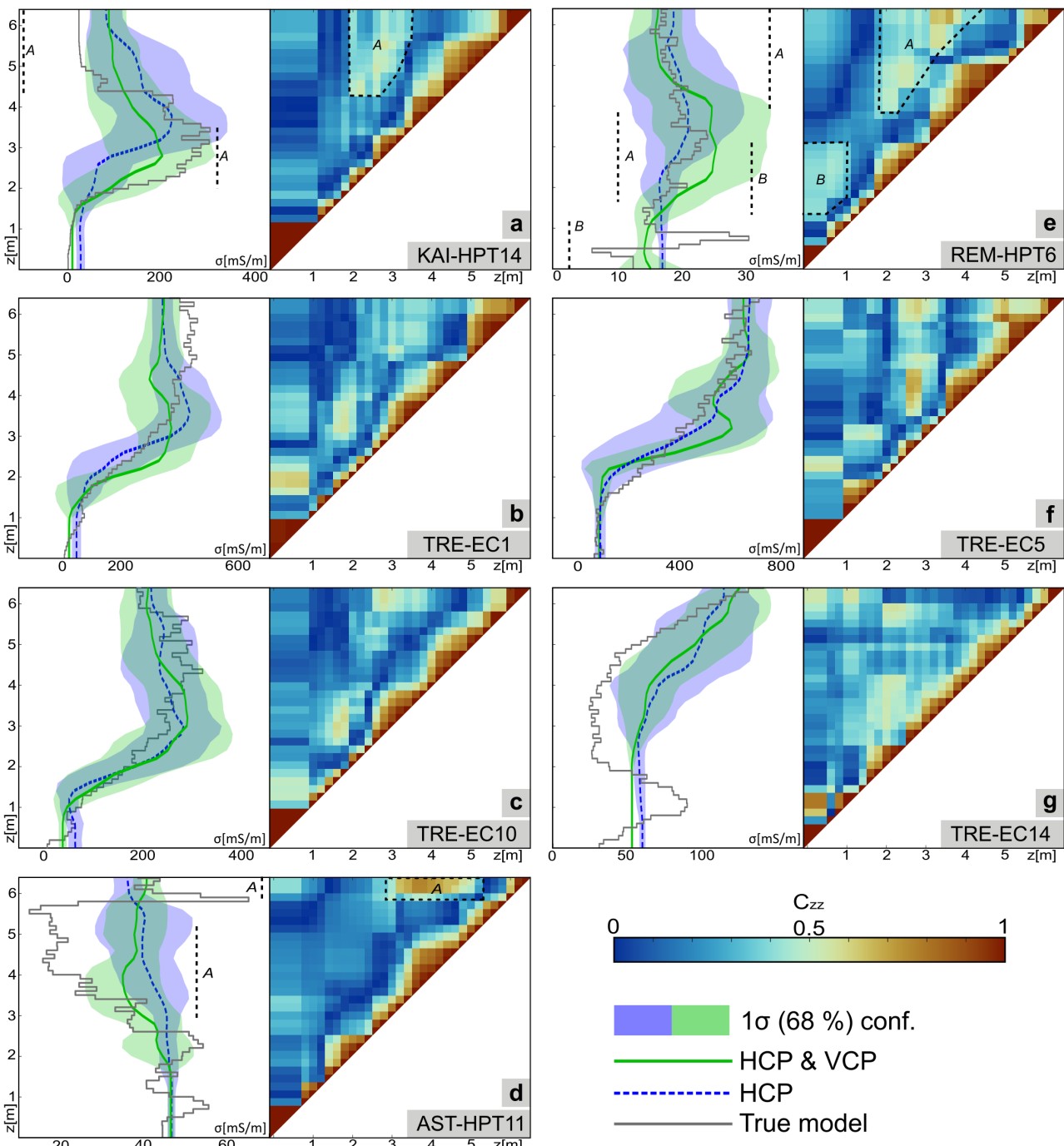

**Figure 5.** Results of DP test models, Part 2: second 7 of the 14 DP test models (**a**–**g**) with original conductivity–depth profile (gray curve), inversion result in terms of expected model for six data points in HCP and VCP orientation (green solid line) and for three data points only in HCP orientation (blue dashed line). The blue and green areas indicate the one $\sigma$ (68%) confidence interval. The gray line represents the true conductivity model. On the right side of each inversion result, the derived covariance matrix is displayed. Labels A and B in (**a**,**d**,**e**) refer to areas of high covariance and are discussed in the text.

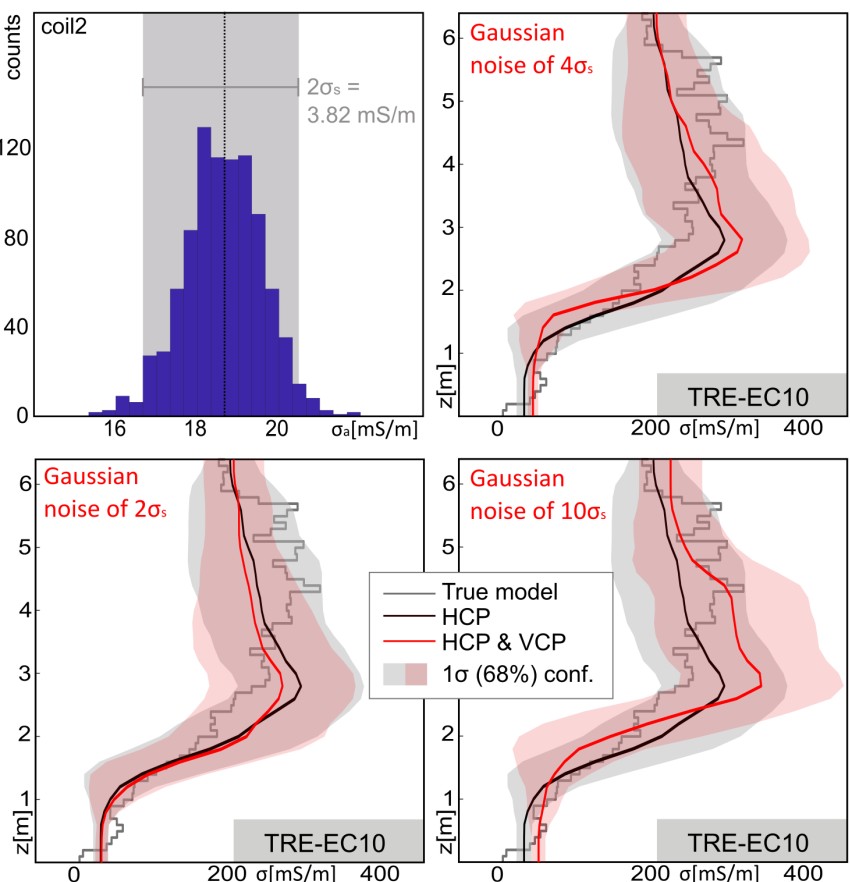

**Figure 6.** Histogram of 1000 measurements with Setup (a) (Figure 3) on the same spot (top left) and inversion results for test model TRE-EC10 with added Gaussian noise having two, four, and ten times the standard deviation of the test measurement. The gray curve represents the true subsurface conductivity model, and red and black curves show the inversion results for both HCP and HCP and VCP data sets with corresponding one $\sigma$ (68%) confidence interval.

The results show that noise only has a negligible effect on the inversion, if in the range of the instrument noise. For larger noise levels that night occur due to external effects, the inversion result does change in parts by more than ten percent (in terms of conductivity of the expected model).

### 3.2. Field Data Applications

After the presentation of inversion results based on modeled data, we now present the results of the three chosen field data experiments, highlighting the feasibility of the inversion approach to different geo-archaeological near-surface settings. Figure 7 shows the result of the CMD Miniexplorer profile of the burial mount example. Profile (a) basically shows a three-layer solution throughout the profile, whereas the lowest layer shows a clear increase in conductivity only in the middle section of the mount, which can be connected with the tomb inside the mount. The profile shows the capability of the inversion procedure to image horizontal changes due to archaeological features using a small offset FDEMI device. The profile furthermore shows good horizontal coherence in terms of the basic layering and the conductivity values.

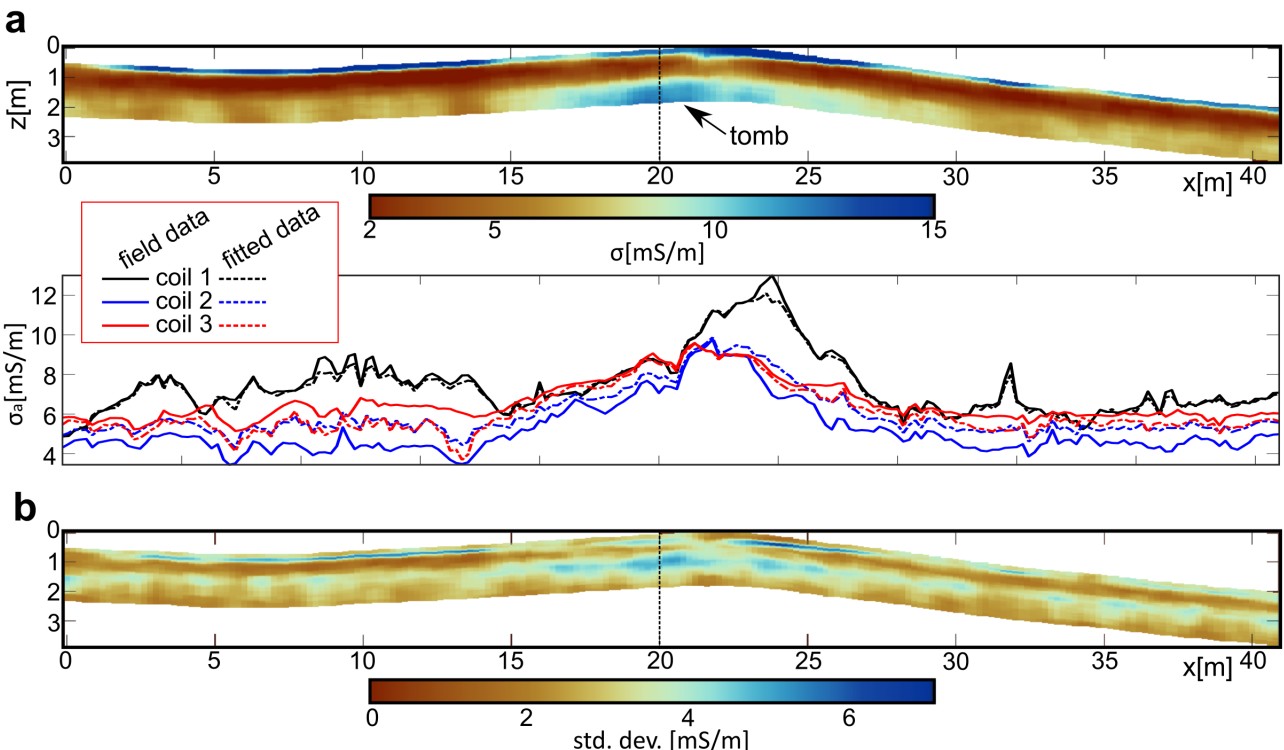

**Figure 7.** Application Example 1: CMD Miniexplorer example of a burial mount, HCP mode, LIN solution forward model. (**a**) Pseudo-2D conductivity profile, horizontally smoothed with a two-model window. Below: resulting data fit with modeled data being based on the expected model for all three coil distances of the CMD Explorer. (**b**) Standard deviation for all resulting models.

Figure 8 shows the result of the CMD Explorer profile from the Roman channel in central Germany. Profile (a) generally shows a two-layer solution, except at the eastern part of the profile, where a three-layer case begins to appear. In the middle of the profile, the top layer shows an increase in conductivity, probably connected to the Roman channel. In that part of the profile, the direct-push EC log has been measured and compared to the EMI inversion result in Figure 8b. The EMI estimated model fits well to the EC-log values, although completely independent in this example. Because of the available ground truth DP data, we chose this example to additionally show the probability density functions (pdf) of conductivities for all layers. The pdf is displayed as a blue-scale plot, showing ambiguity effects in terms of two possible solution maxima at intermediate depths from approximately 1 m to 3 m.

Figure 9 shows the result of the CMD Explorer profile from the coastline of the preboreal lake site. The inverted EMI profile (a) shows no general trend but very different vertical conductivity models, from a nearly constant distribution on the left side of the profile to a two-layer case with high conductivity on top and a two-layer case with a lower conductivity on top. The profile shows a high dynamic range, which can be seen in the plot of several examples of 1D inversion results in (b) extracted from the profile. In comparison to the recorded GPR profile, (c), a clear correlation of the basic layer interfaces can be observed, especially in the left part of the profile.

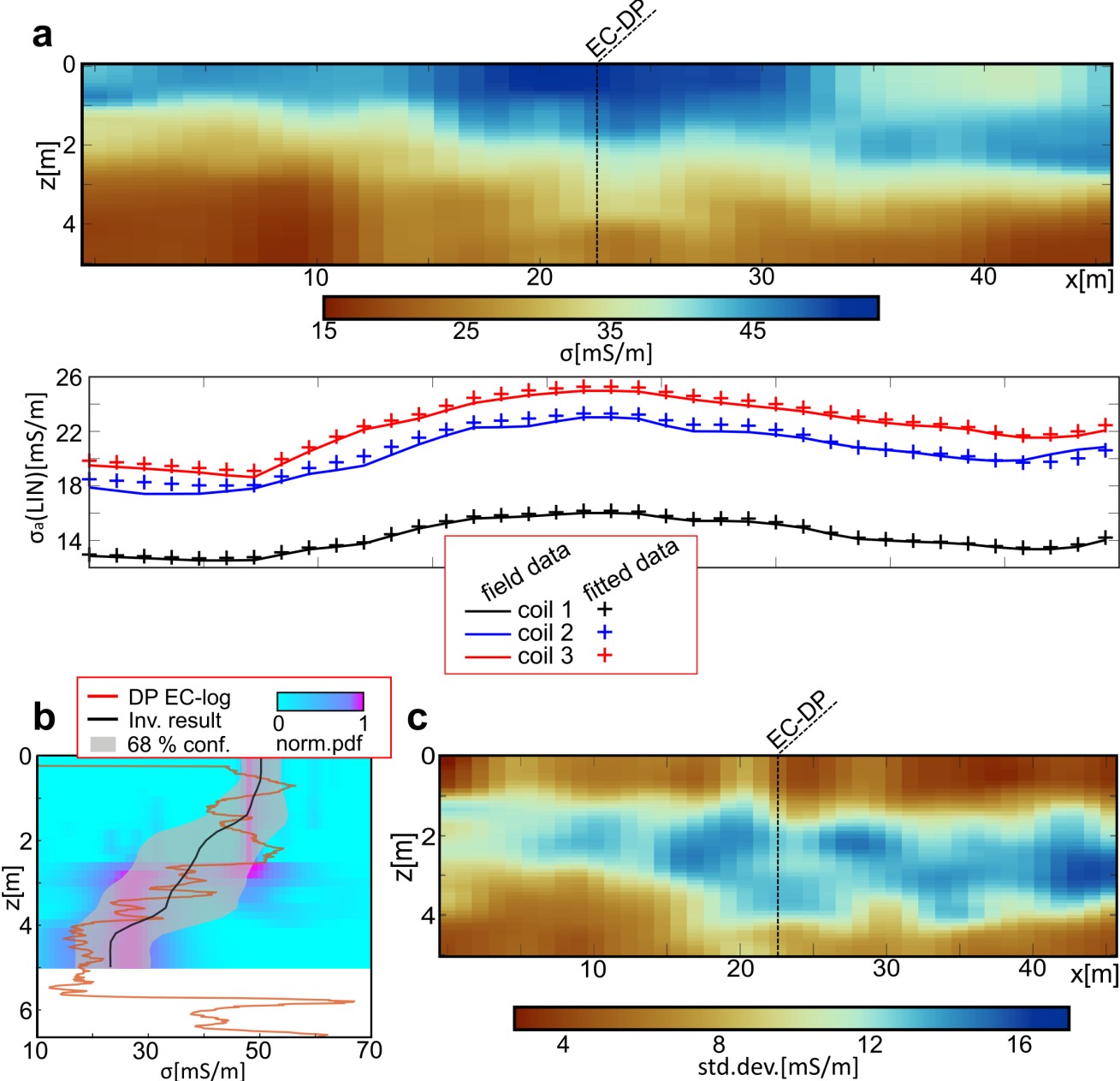

**Figure 8.** Application Example 2: CMD Explorer example compared with EC-log DP data, HCP mode, full solution forward model. (**a**) Pseudo-2D conductivity profile, horizontally smoothed with two-model window. Below: resulting data fit with modeled data being based on the expected model for all three coil distances of the CMD Explorer. (**b**) Example 1D model result with one $\sigma$ (68%) confidence interval and DP EC log in comparison (red). The blue-shaded background shows the probability density functions of all model depths. (**c**) Standard deviation for all resulting models.

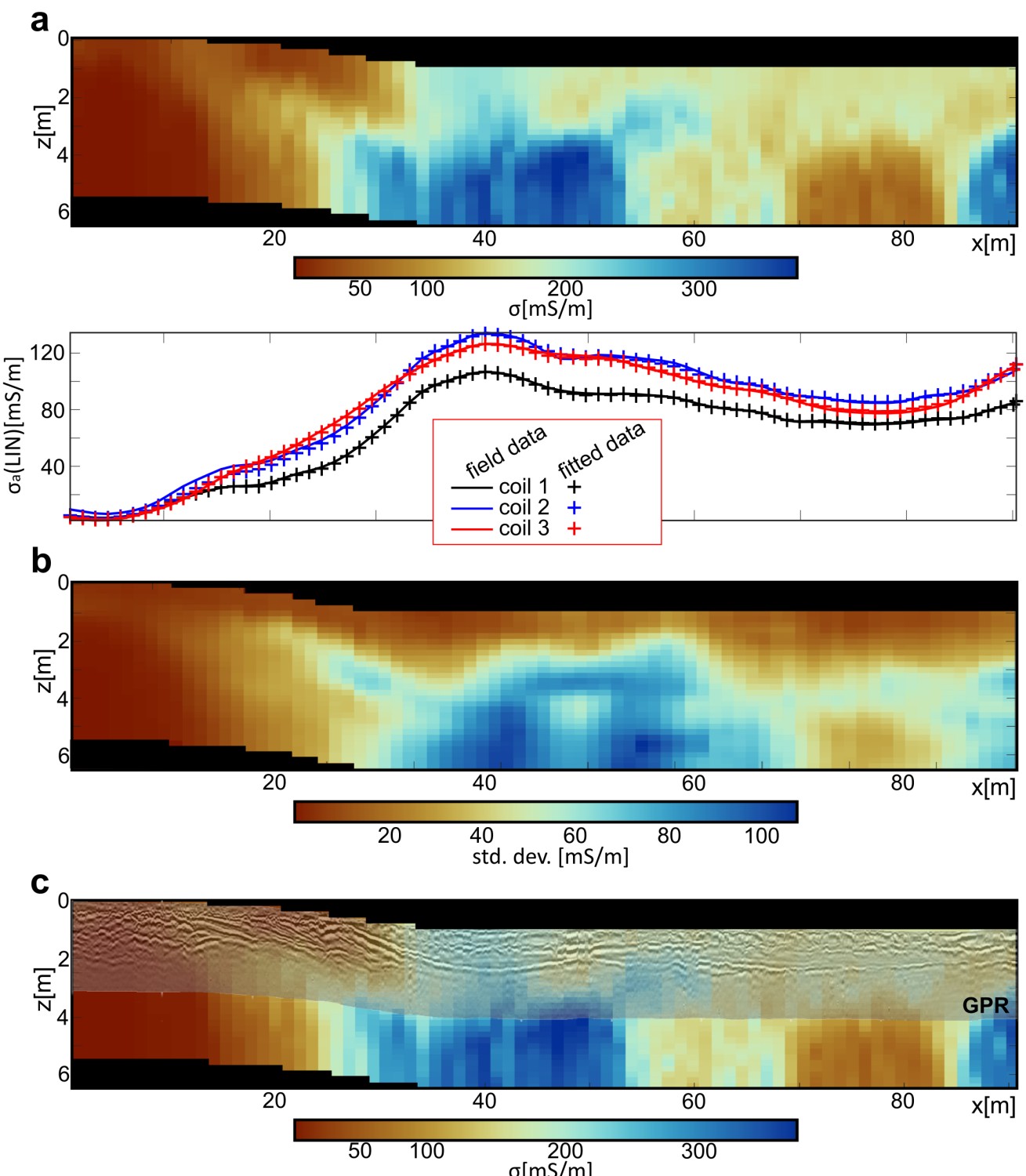

**Figure 9.** Application Example 3: CMD Explorer Example compared with GPR data, HCP mode, full solution forward model, adaptive conductivity parameter space. (**a**) Pseudo-2D conductivity profile. Below: resulting data fit with modeled data being based on the expected model for all three coil distances of the CMD Explorer. (**b**) Standard deviation for all resulting models. (**c**) Comparison of conductivity model with 200 MHz GPR data.

## 4. Discussion

Uncertainty and ambiguity of a geophysical inverse problem are the two key discussion points when evaluating the results of a new inversion scheme. Furthermore, a discussion of the performance of a proposed inversion method is provided in the following section.

The discussion on uncertainty can be based on the results of the presented data examples as well as the broad variety of realistic direct-push test models. As valid for all probabilistic inversion approaches, the introduced method provides a quantitative discussion of parameter estimates and uncertainties due to the large number of forward calculations and misfit evaluations available. An estimated model, variances, and covariance estimates were calculated for all examples. The results on the test models show that the re-sampled estimated models (Figures 4 and 5) resemble the long-wavelength trend of the conductivity–depth models and are thus able to produce a reasonable estimate of the true subsoil situation, especially for the dominant features of the test models (see Spectra in Figure 2). This can also be stated even if only HCP data are available. However, trade-off effects are observed in most of the models, which can be explained by the nature of the EMI method, which is afflicted by equivalent layer problems (e.g., [40]). These effects could be quantified in terms of the covariance matrix and can thus be used for evaluating the result of each single inversion. Besides the stochastic evaluation of model uncertainties and trade-offs, the plausibility of a single-inversion result can also be evaluated by evaluating the coherence of neighboring inversions on a profile. Regarding the presented field data examples it has to be emphasized that the resulting pseudo-2D sections are composed of completely independent 1D inversions with no smoothness constraints. The results in Figures 7–9 show convincing lateral coherency along the profiles, but also reasonable variability to account for changes in geology. The latter can especially be observed in Example 3 (Figure 9).

In order to address the efficiency and performance of the presented method, a comparison to commonly used inversion schemes is provided. We selected two example DP test models (the example KAI-EC6 (Figure 4c) and TEV-EC1 (Figure 4a)) to be inverted with the presented and three other more common inversion methods. Both models were tested for HCP and VCP data, as well as for HCP data only. The compared methods are a traditional global best particle swarm optimization (PSO [41]); an adaption of the Broyden–Fletcher–Goldfarb–Shanno (BFGS) gradient method (L-BFGS-B), as presented in [3]; and the shuffled complex evolution algorithm (SCEUA), also by [3]. As a second population-based method, the PSO inversion was extended by the model-averaging approach presented in Section 2.2 of this paper. The PSO uses the same swarm size and number of iterations as the presented hybrid bee colony approach. Weighting factors were chosen after [7]. The number of free layers in the PSO inversions was fixed to four layers (including half-space) for HCP and VCP data and three-layers for only HCP. Both L-BFGS-B and SCEAU were performed using the software of [3]. These inversions are limited to single-solution models. They were performed both for three-layer models with variable layer depth and conductivity, as well as for six layers of fixed thickness. Both inversions were used with the recommended default settings, but using L1-norm and a smoothing factor of 0.01. Figure 10 shows the result of the comparison. For both models and both measurement configurations, six inversion results are shown. Both the hybrid bee colony and the PSO are presented in averaged models (black and red lines), and L-BFGS-B (blue lines) and SCEUA (green lines) are presented as single-inversion results for both three- (dotted lines) and six-layer models. The results illustrate that the averaging of multiple solution models is superior compared to single-inversion results, as they better represent the stratigraphic features. Figure 10 also shows the differences $\Delta\sigma$ between inversion results and true subsurface models for the two stochastic approaches on the right-hand side of each plot. These differences show that the hybrid bee colony performs better than the PSO (the numbers in the upper right corner are mean differences, black for ABC, red for PSO). This difference is only about 20%, but taking into account that the hybrid bee colony needs

much fewer inversion parameters and frees the researcher from choosing the number of layers to be inverted and the search space bounds, it shows the benefit of the method. In terms of L-BFGS-B and SCEUA, the inversion results are convincing, reconstructing the basic trend of the true models. Nevertheless, these inversions create only single solutions to a multi-modal inversion problem, as illustrated by solutions with different layer numbers. If repeated several times with different starting models and geometries, these methods would probably also enable a successful model averaging. The code presented in this work incorporates all this in one algorithm, enabling a free choice of layer number and combining/averaging very simple to more complex model solutions.

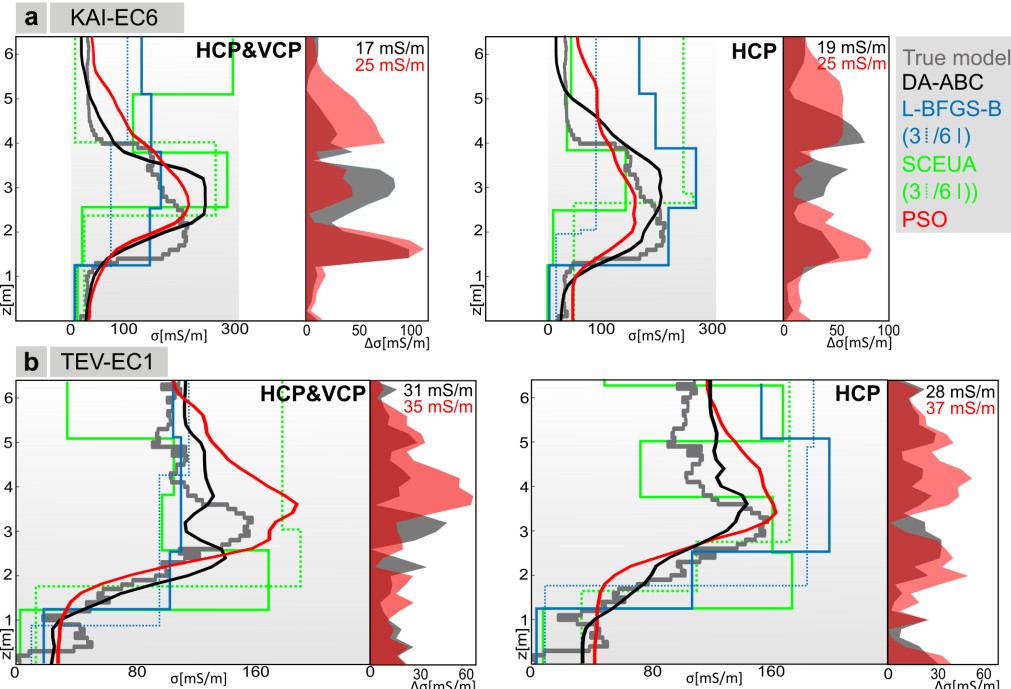

**Figure 10.** Two example inversions taken from Figures 4 and 5. Besides the true subsurface model, the figure shows inversion results for the presented method (black), a basic particle swarm optimization (PSO, red) with model averaging (as presented in the Methods section ), an L-BFGS gradient approach (blue) (implemented in [3]), and en evolutionary algorithm (green) (also in [3]) (**a**). Inversion results for the KAI-EC6 model for both HCP and VCP (left) and only HCP data (right). (**b**) Inversion results for the TEV-EC1 model for both HCP and VCP (left) and only HCP data (right). Both examples also include a difference ($\Delta\sigma$) plot for the comparison of the presented method and PSO.

Besides the general discussion of the method, we address the questions posed in the Introduction. The results clearly show that it is possible to upgrade the bee colony optimization approach with dimension-adapting properties, as used in the MCMC approach by [10] or [25], combining the benefits of the MCMC with swarm intelligence effectiveness. A performance comparison to other inversion methods can be performed by simply checking the number of needed forward calculations. The MCMC by [10] for example used 300,000 forward calculation, respectively. The presented method uses about 120,000 models per run and less, depending on the convergence behavior of each problem. For example, for some test models, only 100 iterations were necessary, reducing the number of forward models to 60.000, which is 20% of the RJ-MCMC. Comparing the lateral coherence of the results on profile- or area-wise inversions can also be of interest. Here, several approaches are mentioned in the literature, e.g., the stitched laterally constrained inversion code proposed by [42] that ties together single 1D inversions along a profile by implementing coherency constraints. The presented inversion method does that without any constraints, exploiting the effect of the average/estimate model. It is thus

not a matter of either finding the one solution agreeing with a measurement that has the smallest possible roughness or finding the one solution that fits best. In the presented case, the inversion creates a smooth (re-sampled) model, statistically representative of all simple solutions found by the inversion, and is thus a more comprehensive way of subsoil conductivity estimate. Beyond that, the presented method frees the user from the choice of solution model. It also needs much fewer parameters than other optimization methods, including number of layers, smoothing parameters, starting model, weighting parameters, and parameter space penalty.

However, it is clear that a 1D inversion scheme suffers from the basic assumption of locally valid 1D subsoil approximations. A true 2D or even 3D inversion based on a 2D or 3D Voronoi discretization, as presented by [25] for seismic tomography, would avoid this problem and also benefit from the presented hybrid optimization scheme. So far, there are some approaches to 3D inversion, but they are either based on simultaneously inverting 1D soundings on a 3D conductivity model that are constrained spatially [43], or based on true 3D deconvolution (e.g., [44]) but restricted to the LIN forward model.

Adapting the dimension of a model can be beneficial in many cases of geophysical inverse problems, especially when the problem is under- or over-determined. The dimension of the problem cannot be directly estimated and needs proper sensitivity analysis. Here, often, the minimum number of layers explaining the data is searched by, e.g., L-curve analysis (e.g., [45]) and used for inversion. This is the case, for example, in spectral analysis of surface waves (e.g., [7]) or vertical electric sounding (e.g., [46]). In these cases, the presented approach could also lead to improvements.

## 5. Conclusions

A dimension-adapting combination of artificial bee colony swarm intelligence optimization and Bayesian reversible jump Monte Carlo Markov chain algorithm has been introduced for the purpose of solving the FDEMI 1D inverse problem. Using artificial bee colony (ABC) optimization solely allows for changing the dimension of the problem throughout the inversion process because the interaction (collective swarm knowledge) of local searches is not based on vector calculations that need the same metric. We showed that the introduction of ABC creates an efficient global searching frame around the local RJ-MCMC approach and thus reduces the number of needed forward calculations significantly. By introducing an averaging approach based on a re-sampling of the numerous simple inversion solutions, the choice of inversion resulting from a set of ambiguous solutions becomes needless. The average or estimated model was proven to give a feasible solution for the dominant long-wavelength trend of all presented test models. These test models represented a broad variation of true subsoil conductivity distributions based on direct-push EC logging from several sites across Europe. The results especially show that even for very sparse data sets of only three values with different coil separations, the basic long-wavelength trend of subsoil conductivity distribution is reconstructed reasonably well. Three field data examples prove the effectiveness of the inversion in terms of lateral model coherence and in comparison to other data sets. The presented optimization approach could easily be adapted for other geophysical problems or even to 2D and 3D problems by using 2D or 3D Voronoi discretization. It thus provides a broad range of applications in future works.

**Author Contributions:** Conceptualization, D.W. and M.M.; methodology, D.W. and M.M.; software, D.W. and M.M.; data curation, P.F., D.W., E.C. and E.E.; writing—original draft preparation, D.W.; writing—review and editing, M.M. and N.P.; visualization, D.W.; project administration, D.W.; funding acquisition, A.V. and D.W. All authors have read and agreed to the published version of the manuscript.

**Funding:** This work was funded by the German Research Foundation (DFG), the Collaborative Research Centre 1266 (Grant 290391021—SFB 1266), the ROOTS Cluster of Excellence (EXC 2150—390870439), and the research project *Archäologische, geoarchäologische und geophysikalische Untersuchun-*

gen zu Eingriffen der Römer ins Fließgewässernetz zwischen Odenwald und Rhein im Bereich des heutigen Landgrabens (Hessisches Ried)-LANDGRABEN (Grant 491982391).

**Data Availability Statement:** The data used in this work can be found with CCBY4 license under https://opendata.uni-kiel.de/receive/fdr_mods_00000017 (accessed on 24 January 2024), https://opendata.uni-kiel.de/receive/fdr_mods_00000018 (accessed on 24 January 2024), https://opendata.uni-kiel.de/receive/fdr_mods_00000019 (accessed on 24 January 2024).

**Acknowledgments:** The authors would like to thank Tina Wunderlich and Aylin Bertram-Striboll for support and fruitful discussions.

**Conflicts of Interest:** The authors declare no conflicts of interest. Author M. Mercker is shareholder of the company Bionum. The remaining authors declare that the research was conducted in the absence of any commercial or financial relationships that could be construed as a potential conflict of interest.

## Abbreviations

The following abbreviations are used in this manuscript:

| | |
|---|---|
| ABC | Artificial Bee Colony |
| FDEMI | Frequency-Domain Electromagnetic Induction |
| HCP | Horizontal Coplanar |
| VCP | Vertical Coplanar |
| DP | Direct Push |
| EC | Electrical Conductivity |
| PSO | Particle Swarm Optimization |
| L-BFGS-B | Limited Memory Broyden–Fletcher–Goldfarb–Shanno |
| SCEUA | Shuffled Complex Evolution Algorithm |

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
