# Peer review of "Artificial Bee Colony Algorithm with Adaptive Parameter Space Dimension: A Promising Tool for Geophysical Electromagnetic Induction Inversion"

_remotesensing, doi:10.3390/rs16030470_

Round 1
Reviewer 1 Report (New Reviewer)
Comments and Suggestions for Authors
This manuscript is needs minor revision.
1- All equations should revise and also units.
2- Some figures resolution should be modified.
3 The manuscript should be revised grammatically well.
4- Advantages and disadvantages should be mentioned.
5- What are the effects of different type of Noise?
Comments on the Quality of English LanguageThe manuscript should be revised grammatically well.
Author Response
Please see the attachment.

Reviewer 2 Report (New Reviewer)
Comments and Suggestions for Authors
The paper is a modification of the RJ-MCMC using the bee colony method. The examples present proved the effects of the method in recovering the resistivity model. However, the main contribution of the method is not highlighted. Here are main my main concerns:
1. what is the purpose to introduce the bee colony method? For the results of the examples presented here, the RJ-MCMC can get a similar results as well.
2. What is the role of the colony method in the MCMC sampling flow? Is it used for re-defining the model space of the prior, or used in the proposal to get the candidate model. After incorporating it, did the detailed balance still can be met?
3. You need to show the advantage/features of your method, especially when it is compared with the existed method.
4. The features of the method also need to be highlighted using your demonstration examples.
5. A strange thing, the data sets did not be shown in the examples both the synthetic and the field ones. Also, how the data are fitted?
6. I see that less number of samples are required in the new method. Can you guarantee the "global" feature of the method? and is the uncertainty estimate still reliable. I think this questions are the central part for a MCMC method.
Comments on the Quality of English LanguageNo specific comments on this.
Author Response
Please see the attachment.

Reviewer 3 Report (New Reviewer)
Comments and Suggestions for Authors
Below are some detailed comments:
1. As a novel stochastic optimization method, the authors did not give enough details to reveal the features of the proposed method. I recommend the authors to address the merits of the proposed method over the conventional RJ-MCMC method, especially highlighting them with the examples.
2. The EMI is very similar to the CSEM method, please illustrate of briefly introduce the difference between these two types of method.
3. in 2.1 the Forward problem, what you want to show in this section? I think you did a good job in introducing the LIN approximation here, not the forward problem.
4. line 131: what is the code here?
5. can you discuss the reason that you use the expected model as the results?
6. Figure 4: How do you think about the confidence interval here? It really shows a large region and attaches large uncertainty on the results, and this is the reason that I want you to discuss the expected model.
7. line 351: Based on the following analysis, the examples below need to be numbered.
8. line 352: I think a chart to demonstrate and compare the three examples is better.
9. For the section of field example (such as figure 6a), it would be good to add the section of the confidence interval, as you have actually get the interval at each recording station.
10. For the field example: you need to add the plots of the raw data and figures to show how the data are fitted.
11. Figure 8, you attached the results of the GPR, but the results were not cited and discussed in the main texts.
Comments on the Quality of English Language1. The first paragraph of the INTRUDCTION section is way too long and several topics are included in it.
2. I Suggest using short sentences.
Round 2
Reviewer 2 Report (New Reviewer)
Comments and Suggestions for Authors
I think the authors did a good job in revising the manuscript according to my comments. No further comments this time.
Author Response
REVIEWER 2
I think the authors did a good job in revising the manuscript according to my comments. No further comments this time.
Thank you again for the time and effort that you dedicated to our manuscript.
Reviewer 3 Report (New Reviewer)
Comments and Suggestions for Authors
The authors have made modifications based on my concerns. I think the authors overall did a good job. Several additional comments:
1. The tracks of the modifications have not been kept, which makes it difficult for the reviewer to follow the modifications. I think it is necessary to upload the manuscript with track changes for a revision.
2. Still, for the first paragraph of the introduction, the authors introduced several topics, leading to a real long paragraph. It would better if it can be divided into several paragraphs.
3. For a MCMC inversion, I would like to see the posteriors that can be used to quantify the uncertainty of the inversion results.
Author Response
REVIEWER 3
Thank you again for the time and effort that you dedicated to our manuscript. Here is a point-by-point response to your comments and concerns:
The authors have made modifications based on my concerns. I think the authors overall did a good job. Several additional comments:
1. The tracks of the modifications have not been kept, which makes it difficult for the reviewer to follow the modifications. I think it is necessary to upload the manuscript with track changes for a revision.
RESPONSE: We have to apologize for the inconvenience. We would definitely have uploaded a tracked version of the manuscript, but unfortunately, the manuscript was written in LaTeX/Overleaf, which does not provide a highlight changes tool. We have added a version of the manuscript as supplemental material where we highlighted all text changes.
2. Still, for the first paragraph of the introduction, the authors introduced several topics, leading to a real long paragraph. It would better if it can be divided into several paragraphs.
RESPONSE: Yes, the sweeping EMI and Inversion introduction is a too long block. We have devided that part into four paragraphs, optically seperating EMI introduction, EMI inverse problem, stochastic inversion methods, and aims of the present work.
3. For a MCMC inversion, I would like to see the posteriors that can be used to quantify the uncertainty of the inversion results.
RESPONSE: Thank you for your remark. Unfortunately, this is not possible, since we use too few MCMC steps in the context of dimension adaption in order to visualise any posterior here:
The posterior, provided by a MCMC would in general give us an updated measure of the posterior probability density for the model parameters with evolving iteration. This surely would provide an uncertainty measure for the single, blocked solution (base or "simple" models in Figure 1) at each food source of the ABC.
However, MCMC is only a sub-part of the ABC inversion (section 2.4, point 7a, item 2). It is used to make the Reversible Jump decisions and is only executed a few times for each food source (which is why we used this Bayesian approach), probably resulting in weakly sampled pdf.
The measure of uncertainty used in the paper (section 2.2) uses the variance of all found base/simple solution models to estimate the uncertainty, like in Ryberg & Haberland. However, as we are assuming a Boltzman distribution to weight the single solutions by misfit, a comparison to the posterior probability density function really would be helpful to understand if this assumption is valid and to have a second opportunity to evaluate ambiguity. The best way to do that, based on the ABC frame of the inversion, is to calculate a distribution für each sublayer in the resampled models. Furthermore, we observed that the inversion tends towards simple solutions, due to the physics of the EMI method and the sparse data available. A posterior pdf would show that. So we have added an example pdf in the application example 2, where DP-data as true subsurface conductivity distribution is available. The example reveals ambiguity and a trend towards different two layer solutions. Nevertheless, in terms of parameter uncertainty, the general approach as used in the paper is comparable.
This manuscript is a resubmission of an earlier submission. The following is a list of the peer review reports and author responses from that submission.
Round 1
Reviewer 1 Report
Comments and Suggestions for Authors
The article titled "Artificial Bee Colony Algorithm with Adaptive Parameter Space Dimension: A Promising Tool for Geophysical (EMI) Inversion" presents an innovative approach to geophysical inversion using the Artificial Bee Colony (ABC) algorithm with an adaptive parameter space dimension. The study holds significant promise for enhancing the accuracy and efficiency of Electromagnetic Induction (EMI) inversion techniques, and its findings have noteworthy implications for geophysical exploration. The authors skillfully introduce the fundamental challenges in geophysical inversion, highlighting the necessity for robust optimization techniques to overcome the inherent non-linearity and ill-posedness of the problem.
However following questions need to be addressed before publication.
1- The article provides an extensive mathematical explanation for the modeling; however, the models introduced do not adequately support the mathematical framework. While the intention to describe two distinct sites (Site 13 and Site 3) is stated, there is a need for the author to establish a stronger justification for the chosen model representations or consider streamlining irrelevant sections.
2- Here it is noteworthy that the trade-off with the deeper higher 410 conductive layer, as observed in Figure 5 does not appear in the EMI inversion as shown 411 later. This is due to the fact, that EMI device was carried at 1 m height (see Figure ??), 412 reducing the overall depth range, and thus not reaching this conductivity outlier at about 6 413 m depth.
The above statement is completely confusing and there is no figure is mentioned as well as highlighted.
3- Within the section detailing the field data experiment, the immediate presentation of results holds significance. However, it is advisable to preface these results with a couple of paragraphs that establish a coherent connection between the modeling phase and the field data experiment. Such an approach is vital to preserve the thematic unity of the article, as without the linkage between these two sections, they risk appearing disparate and disjointed.
4- The discussion section references codes; however, no corresponding details about these codes are provided within the text. While the inclusion of the actual codes might not be feasible, it is suggested to elucidate the foundational principles or concepts underlying the codes. Furthermore, the absence of any description regarding the field data necessitates attention to ensure its inclusion and proper coverage in the text.
Comments on the Quality of English LanguageMinor editing of English language is required.
Author Response
On behalf of all authors, I would like to thank the referee for the helpful and constructive comments. My answers and implementation can be found directly underneath each comment.
Comments and Suggestions for Authors
The article titled "Artificial Bee Colony Algorithm with Adaptive Parameter Space Dimension: A Promising Tool for Geophysical (EMI) Inversion" presents an innovative approach to geophysical inversion using the Artificial Bee Colony (ABC) algorithm with an adaptive parameter space dimension. The study holds significant promise for enhancing the accuracy and efficiency of Electromagnetic Induction (EMI) inversion techniques, and its findings have noteworthy implications for geophysical exploration. The authors skillfully introduce the fundamental challenges in geophysical inversion, highlighting the necessity for robust optimization techniques to overcome the inherent non-linearity and ill-posedness of the problem.
However following questions need to be addressed before publication.
1- The article provides an extensive mathematical explanation for the modeling; however, the models introduced do not adequately support the mathematical framework. While the intention to describe two distinct sites (Site 13 and Site 3) is stated, there is a need for the author to establish a stronger justification for the chosen model representations or consider streamlining irrelevant sections.
ANSWER:I do not fully understand this point. The term model is used in the text for tupel of parameter space consisting of layer depths, layer conductivities and dimensionality of the tupel itself (Formula 4). They are used thoughout the mathematical formulation of the inversion approach referred to as m (vector). The model representation as decribed in section 2.2 is of course only one possible way to dicretize layered subsurface models. Other approaches would be
-A stack of several small layers with fixed thickness as done by e.g. Constable 1987 (as cited in section 2.2); with smoothing contraints.
-A functional (e.g. spline, polynomial functions) representation.
As the first mentioned approach would lead to a larger number of unknowns, we withdrew this method. The second approach is problematic in terms of the MCMC method, because the birth and death step is difficult to accomplish, if e.g. polynomial coefficients are model parameters; leading to quite large changes in the model representation.
2- Here it is noteworthy that the trade-off with the deeper higher 410 conductive layer, as observed in Figure 5 does not appear in the EMI inversion as shown 411 later. This is due to the fact, that EMI device was carried at 1 m height (see Figure ??), 412 reducing the overall depth range, and thus not reaching this conductivity outlier at about 6 413 m depth.
The above statement is completely confusing and there is no figure is mentioned as well as highlighted.
ANSWER:I have changed the text according to your comment, following line 414 in the revised manuscript.
3- Within the section detailing the field data experiment, the immediate presentation of results holds significance. However, it is advisable to preface these results with a couple of paragraphs that establish a coherent connection between the modeling phase and the field data experiment. Such an approach is vital to preserve the thematic unity of the article, as without the linkage between these two sections, they risk appearing disparate and disjointed.
ANSWER:I have added a short text passage, trying to establish a connection between the two paragraphs (following line 422).
4- The discussion section references codes; however, no corresponding details about these codes are provided within the text. While the inclusion of the actual codes might not be feasible, it is suggested to elucidate the foundational principles or concepts underlying the codes. Furthermore, the absence of any description regarding the field data necessitates attention to ensure its inclusion and proper coverage in the text.
ANSWER:That is indeed misleading, thank you. I have changed the phrase 'code' to 'inversion method', because the text speaks of the inversion method presented in the paper, not the code itself, of course. I have also changed the text discussing the field data examples according to your comment.
Reviewer 2 Report
Comments and Suggestions for Authors
The manuscript is presented in 30 pages with 45 references, 8 figures without tables. It has valuable research in electromagnetics. Authors present a novel 1D stochastic optimization approach that combines dimension adapting Reversible Jump Markov Chain Monte Carlo (MCMC) with Artificial Bee Colony (ABC) optimization for geophysical inversion. As bees we consider Hired Drones, Jumper Drones, and Exploration Drones. Authors present algorithm for approach with examples of EC-logs using a Geoprobe 540 MO system mounted to a Nordmeyer drill rig in combination with a Geoprobe K6050 HPT probe or a Geoprobe SC400 Conductivity Wenner EC Probe. (measured in horizontal and vertical orientation. These test models represented a broad variation of true subsoil conductivity distributions based on direct push EC logging from several sites across Europe. So I prefer to add it in the abstract (that field data is made in some sites in Europe of archaeological value).
Most of my comments are about stylistic and grammar.
1 blank page is excessive
Please, add zip codes to metadata
Line 9 abstract. field data – please, specify sites or instrumentation
Please, follow journal’s guidelines. The first paragraph should be not intended. However, following paragraphs are intended. Also, please keep space after the figure captions as in layout.
Line 104 ?? blank?
Line 152 Nz = int(). Is it an integral? Please, write as normal, not in programming language. The same for the line 385 ϵ1 = 1.0e−4 mS/m, e is using for programming as 10.
Line 161. Full- or LIN forward solutions. Please, write it properly. Sometimes you use names as full, FULL (figures 7, 8) of Full forward solutions. You should be respectful for readers and give brief information what is it and the differences between these two. full solution, refers to the full solution of Maxwell's equation. The LIN approximation proposed [27] …
Line 234 with r (in italic font)
Line 254 (c) check … all foodsources (as you use everywhere written in one word)
Eq. 11. dQ which indexes does it have? c, j?
Line 266. Point 9. constant ϵ2.Why ϵ2 not ϵ1? Because, further you mention only limitations for constant ϵ1. Look in line 325.
Eq. 13 what is b in Nb?
Please, keep space between values and units, e.g. 5 mS/m. 400 MHz
Line 325-364 number of the figure is missed Figure ??
Line 359 what is burgi? Areas? It is in Latin.
Line 367 gyttja layers. What is gyttja? Mud?
Line 397 if you mention Figures 4 and 5, you should put figures right away after this paragraph. As it is difficult to understand, what are these models. I also ask you to explain why HPT 1, 2, 6, 11 (what is the difference?) EC1 & 10.
Line 412 Figure ??
Figures 4, 5 please add in caption a…g BIE-HPT2,DUV-HPT1 and TRE-EC1 & 10, REM-HPT6, BIE-HPT2, AST-HPT11… as the indications are too small to observe it clearly.
forward calculation and iterations are the same?
Line 479 LCI code. Please, use full name, as you use it the first time Laterally constrained inversion
Line 493 3D Voronoi
Line 496 1D soundings on a 3D conductivity
Line 501 please, give a reference, e.g. The Use of the L-Curve in the Regularization of Discrete Ill-Posed Problems // 1993 SIAM Journal on Scientific Computing 14(6):1487-1503 DOI: 10.1137/0914086
L. 524-531. Please, delete this text. please use the CRediT taxonomy for the authorship explanation.
In references please use full surnames for second and further authors, not Duram, H.; A.B.C.D.E.;
[1] … Electromagnetic methods in applied geophysics,
Please, delete doubles https://doi.org/https://doi.org/
[22] 19(21), 4753,
[30] in German?
[34] 12(6), 245,
[35] 10(8), 314
[39] Available at... Accessed on
[40] Ancient Near Eastern Studies 2019, ??, ??
[45] 79th EAGE Conference and Exhibition 2017, 2017, 1–3.
If it possible, could you provide a library with codes/scripts for this research? Or mention which software or product did you use.
Comments on the Quality of English LanguageEnglish is fine. However, there some typos. See my comments.
Author Response
Thank you very much for your constructive and helpfull comments. My answers and implementation can be found directly underneath each comment.
The manuscript is presented in 30 pages with 45 references, 8 figures without tables. It has valuable research in electromagnetics. Authors present a novel 1D stochastic optimization approach that combines dimension adapting Reversible Jump Markov Chain Monte Carlo (MCMC) with Artificial Bee Colony (ABC) optimization for geophysical inversion. As bees we consider Hired Drones, Jumper Drones, and Exploration Drones. Authors present algorithm for approach with examples of EC-logs using a Geoprobe 540 MO system mounted to a Nordmeyer drill rig in combination with a Geoprobe K6050 HPT probe or a Geoprobe SC400 Conductivity Wenner EC Probe. (measured in horizontal and vertical orientation. These test models represented a broad variation of true subsoil conductivity distributions based on direct push EC logging from several sites across Europe. So I prefer to add it in the abstract (that field data is made in some sites in Europe of archaeological value).
I have changed the abstract according to your advice.
Most of my comments are about stylistic and grammar.
1 blank page is excessive
ANSWER:this was somehow added by the MDPI LaTeX template and I have no idea how to remove it
Please, add zip codes to metadata
ANSWER:zip codes were added
Line 9 abstract. field data – please, specify sites or instrumentation
ANSWER:I have added the sites and the Direct Push and EMI information in the abstract
Please, follow journal’s guidelines. The first paragraph should be not intended. However, following paragraphs are intended. Also, please keep space after the figure captions as in layout.
ANSWER:This is again the LaTeX template by MDPI, but I have changed that manually.
Line 104 ?? blank?
ANSWER:Yes
Line 152 Nz = int(). Is it an integral? Please, write as normal, not in programming language. The same for the line 385 ϵ1 = 1.0e−4 mS/m, e is using for programming as 10.
ANSWER:I have removed the integer cast and both e^
Line 161. Full- or LIN forward solutions. Please, write it properly. Sometimes you use names as full, FULL (figures 7, 8) of Full forward solutions. You should be respectful for readers and give brief information what is it and the differences between these two. full solution, refers to the full solution of Maxwell's equation. The LIN approximation proposed [27] …
ANSWER: I introduced the two phrases in the section Forward Problem.
Line 234 with r (in italic font)
ANSWER:changed
Line 254 (c) check … all foodsources (as you use everywhere written in one word)
ANSWER:checked and changed
Eq. 11. dQ which indexes does it have? c, j?
ANSWER:j, I have added the index
Line 266. Point 9. constant ϵ2.Why ϵ2 not ϵ1? Because, further you mention only limitations for constant ϵ1. Look in line 325.
ANSWER:e1 is the foodsource stagnation constant, e2 is the general stopping criterion. I added the e2 information to the paragraph at line 325
Eq. 13 what is b in Nb?
ANSWER:This wasn't introduced; this is the number of models used for the statistical analysis. I have added that to the text.
Please, keep space between values and units, e.g. 5 mS/m. 400 MHz
ANSWER:Some were missing; changed that
Line 325-364 number of the figure is missed Figure ??
ANSWER:The figure dissappeared in the final manuscript. I have checked that.
Line 359 what is burgi? Areas? It is in Latin.
ANSWER:Burgi is the plural of Burgus. I changed it to fortlets
Line 367 gyttja layers. What is gyttja? Mud?
ANSWER:yes, mud of organic origin deposited in lakes and bogs; I added that
Line 397 if you mention Figures 4 and 5, you should put figures right away after this paragraph. As it is difficult to understand, what are these models. I also ask you to explain why HPT 1, 2, 6, 11 (what is the difference?) EC1 & 10.
ANSWER:sure, I moved the figures. HPTx and ECx are the field names of the logs as used in the corresponding articles where these curves are published. I decided to keep the names to make it easier to connect them.
Line 412 Figure ??
ANSWER:same problem with figure 3 as above; solved
Figures 4, 5 please add in caption a…g BIE-HPT2,DUV-HPT1 and TRE-EC1 & 10, REM-HPT6, BIE-HPT2, AST-HPT11… as the indications are too small to observe it clearly.
ANSWER: I added the letters and changed the text accordingly. I had to change the figures because there was an error in the display of the 95% confidence intervals which were far too small.
forward calculation and iterations are the same?
ANSWER:no, forward calculations are either LIN or full solutions of the forward problem; iterations are the iterations of the inversion. I couldn't find the paragrapth where this might be misleading
Line 479 LCI code. Please, use full name, as you use it the first time Laterally constrained inversion
ANSWER:changed
Line 493 3D Voronoi
ANSWER:changed
Line 496 1D soundings on a 3D conductivity
ANSWER:changed
Line 501 please, give a reference, e.g. The Use of the L-Curve in the Regularization of Discrete Ill-Posed Problems // 1993 SIAM Journal on Scientific Computing 14(6):1487-1503 DOI: 10.1137/0914086
ANSWER:added
L. 524-531. Please, delete this text. please use the CRediT taxonomy for the authorship explanation.
ANSWER:changed
In references please use full surnames for second and further authors, not Duram, H.; A.B.C.D.E.;
ANSWER:a lot of ands were missing in bibtex; I changed that
[1] … Electromagnetic methods in applied geophysics,
ANSWER:changed
Please, delete doubles https://doi.org/https://doi.org/
ANSWER:changed
[22] 19(21), 4753,
ANSWER:checked
[30] in German?
ANSWER:yes, as far as I know, there is no translation of that book.
[34] 12(6), 245,
ANSWER:changed
[35] 10(8), 314
ANSWER:changed
[39] Available at... Accessed on
ANSWER:changed
[40] Ancient Near Eastern Studies 2019, ??, ??
ANSWER:checked
[45] 79th EAGE Conference and Exhibition 2017, 2017, 1–3.
ANSWER:checked
If it possible, could you provide a library with codes/scripts for this research? Or mention which software or product did you use.
ANSWER:The codes will be available on GitHub soon. I will add that information during the publication process. The code is written in Matlab
Reviewer 3 Report
Comments and Suggestions for Authors
1. Introduction and methodological principles sections are verbose and should be appropriately streamlined.
2. The paper mentions, "However, in many cases the optimization problems are highly non-unique, making the application of global searching probabilistic methods worthwhile." Are the case studies in this paper considered "highly non-unique"? Can you provide a comparison with traditional gradient-based methods?
3. The paper mentions, "While these algorithms are generally fast and provide useful estimates of subsurface properties, an analysis of parameter uncertainty, correlation, and non-uniqueness is often left unaddressed." Is this conclusion still valid today, or has there been progress in addressing these issues over the past 12 years?
4. The paper mentions, "As most stochastic methods like Evolutionary Algorithms/Strategies (e.g. [30]), the Neighborhood Algorithm (NA, [31]), or Particle Swarm Optimization (PSO) and its Variants (e.g. [9]) use the metric of parameter space to recombine particles or individuals or agents, a dimension-adapting strategy is thus impossible to implement. Different particles of different dimensions cannot be combined and have no unique distance measure. All these algorithms are thus not suitable to be used as a global searcher if the dimension of a problem is a parameter, too." Please provide a comparison with other evolutionary and probabilistic algorithms to support this claim in the paper.
5. The paper lacks explanations for many parameter settings in ABC and MCMC. Please provide additional information, and consider conducting sensitivity analysis to ensure the rationality of these parameter choices.
Author Response
On behalf of all authors, I would like to thank the referee for the helpful and constructive comments. My answers and implementation can be found directly underneath each comment.
1. Introduction and methodological principles sections are verbose(langatmig) and should be appropriately streamlined.
ANSWER:I have checked the text for redundancies and verbose parts. Furthermore I have merged the model parametrization and estimated model calculations sections to one section.
2. The paper mentions, "However, in many cases the optimization problems are highly non-unique, making the application of global searching probabilistic methods worthwhile." Are the case studies in this paper considered "highly non-unique"? Can you provide a comparison with traditional gradient-based methods?
ANSWER:As the code is a dimension adapting approach, a direct comparison with gradient methods (e.g. Newton-like methods; which are by definition bound to a mapping from |R^N to |R ) is not possible. Due to the problem of equivalent layers in EMI (e.g.) and the low number of measurement values (3 to 6, depending on the EMI configuration), leading to an underestimated problem, one can speak of a highly non-unique problem. This is addressed in the paper by the covariance matrices provided with the test model inversions. Traditional gradient based methods are highly effective (if Newton-like) in a convex parameter space (q-superlinear convergence, e.g. P. Kosmol 1993 - ISBN 3-519-12085-2), however they are always bound to the convex area of parameter space around the starting model.
3. The paper mentions, "While these algorithms are generally fast and provide useful estimates of subsurface properties, an analysis of parameter uncertainty, correlation, and non-uniqueness is often left unaddressed." Is this conclusion still valid today, or has there been progress in addressing these issues over the past 12 years?
ANSWER:Yes it is. There are several approaches to address uncertainty and trade-off effects. In terms of the countless stochastic inversion approaches on different geophysical problems, the solution is easy due to the large number of available misfit evaluations and large parameter spaces, as mentioned and referred to in the text. When using gradient based inversion methods, adressing those parameters is more difficult.
A single gradient search is bound to its convex surrounding of the starting model (which it cannot leave by search steps) and thus cannot provide a global uncertainty measure. However, fixing all but one model parameters, the uncertainty of the found minimum and the confinement of single parameters can be estimated (e.g. Bohlen et al. 2004).
If however, the gradient search is performed at several starting points, leading to different solution models, statistical analysis can be applied as well. This on the other hand, if globally guided, is a hybrid inversion approach incorporating stochastic inversion on top of effective local gradient searchers (e.g. Chunduru et al. 1997).
(Hybrid optimization methods for geophysical inversion Raghu K. Chunduru, Mrinal K. Sen, and Paul L. Stoffa GEOPHYSICS 1997 62:4, 1196-1207)
(Bohlen T., Kugler S., Klein G., Theilen F., 2004. 1.5-D inversion of lateral variation of Scholte wave dispersion, Geophysics, 69, 330–344.)
4. The paper mentions, "As most stochastic methods like Evolutionary Algorithms/Strategies (e.g. [30]), the Neighborhood Algorithm (NA, [31]), or Particle Swarm Optimization (PSO) and its Variants (e.g. [9]) use the metric of parameter space to recombine particles or individuals or agents, a dimension-adapting strategy is thus impossible to implement. Different particles of different dimensions cannot be combined and have no unique distance measure. All these algorithms are thus not suitable to be used as a global searcher if the dimension of a problem is a parameter, too." Please provide a comparison with other evolutionary and probabilistic algorithms to support this claim in the paper.
ANSWER:A comparison to these algorithms is not possible, because they rely on a metric, combining (recombination in EA, Movement in PSO) model vectors in terms of binary operations (summation, subtraction,...). These operations are not possible if the two vectors have different lengths (dimensions). I rephrased the paragraph to make that more clear.
5. The paper lacks explanations for many parameter settings in ABC and MCMC. Please provide additional information, and consider conducting sensitivity analysis to ensure the rationality of these parameter choices.
ANSWER:In the section DP Testmodels the most important parameters were addressed for the test inversions. Actually e2 and sigma2 were missing in the text and were added accordingly.
Round 2
Reviewer 3 Report
Comments and Suggestions for Authors
The author cannot definitively assess the efficacy of the algorithm proposed for addressing "highly non-unique optimization problems" or determine the practical value of employing a global search probability method. The analysis of the shortcomings of traditional algorithms remains at a level dating back to over a decade ago, without a comparative analysis with contemporary advanced algorithms (with no references to articles published in the past three years). Furthermore, there is no comparison made with traditional heuristic algorithms like PSO (Particle Swarm Optimization) or simulated annealing, which makes the motivation behind this study insufficient and challenging to establish the necessity of the proposed algorithm for EMI.
This paper may be better suited for journals focused on heuristic algorithms, as it represents an improvement and effective application of the proposed artificial bee colony algorithm.